# Analyzing Determinants of Job Satisfaction Based on Two-Factor Theory

Byunghyun Lee [1], Changjae Lee [2], Ilyoung Choi [1] and Jaekyeong Kim [1,3,*]

1  Department of Big Data Analytics, Kyung Hee University, 26 Kyungheedae-ro, Dongdaemun-gu, Seoul 02453, Korea
2  Department of Business Administration, Graduate School, Kyung Hee University, 26 Kyungheedae-ro, Dongdaemun-gu, Seoul 02453, Korea
3  School of Management, Kyung Hee University, 26 Kyungheedae-ro, Dongdaemun-gu, Seoul 02453, Korea
*  Correspondence: jaek@khu.ac.kr; Tel.: +82-2-961-9355

**Abstract:** For one company to have a competitive advantage and sustainability over others, its human resource management is of the utmost importance to secure competent employees. As job satisfaction plays a critical role in securing excellent manpower and enhancing corporate performance, it is essential to identify factors that would affect employees' job satisfaction. Recently, writing reviews with integrity on job portal sites by former and current employees has become prevalent as such websites have guaranteed the reviewers' anonymity. For this reason, we collected a vast amount of review data over nine industries, such as IT web communication, from one of the representative job portal sites in South Korea, Job Planet, and investigated factors that affect one's job satisfaction based on the two-factor theory. As a result, it was found that (1) both motivation and hygiene factors had a substantial effect on job satisfaction over all industries; (2) the moderating effect between former and current employees was different for each industry; and (3) there was no moderating effect on job satisfaction between motivation and hygiene factors.

**Keywords:** two-factor theory; job satisfaction; job portal site; review data



## 1. Introduction

With the advent of the 4th industrial revolution, companies have leveraged digital transformation to secure their competitive advantage and sustainability. However, according to one report by McKinsey, about 70% of companies who attempt to leverage digital transformation will fail as they tend to have a mere focus on technology adoption than their employees [1]. In other words, although employees are ones who lead innovation and change, companies often perceive them as the means of production. However, for a company to secure a competitive advantage and business sustainability, it is necessary to manage their human resources in an effective manner through retention of competent employees within the organization [2,3]. If one competent employee were to leave his or her company, the competitive power of the firm would deteriorate through the loss of accumulated knowledge or its technical skills [4,5].

As effective management of employee turnover is required, research has been steadily conducted to analyze employees' turnover intention and behaviors. According to previous studies, job satisfaction has been identified as the primary factor that would lower turnover intention [4–7]. In fact, job satisfaction plays a critical role in corporate operation because it not only affects employees' turnover intention but also other factors such as job commitment and performance [8,9]. Since job satisfaction is crucial in securing human resources and thereby creating better corporate performance, it is vital to identify factors that influence employees' job satisfaction.

Based on the two-factor theory, many previous studies have figured motivation and hygiene factors as ones that greatly affect job satisfaction [10–14]. However, using a

survey may cause a social desirability bias, in which people do not want to present their specific opinions due to an underlying concern of possible disadvantages that they may encounter [15].

Until nowadays, there has been a proliferation of job portal sites which not only provided a company's reviews and ratings written by former and current employees, but also other company-related information such as welfare benefits, salaries, and interview questions. One of the main reasons for such proliferation comes from the protection of anonymity of all users, which have allowed them to freely express their opinions and actively share information. Relatively speaking, since there is higher tendency of employees using job portal sites to provide personal yet specific experiences and opinions [16], it is presumable that they would not yield socially desirable answers. Therefore, there has been ongoing research on job satisfaction using vast amounts of data collected from job portal sites [17–20]. In addition, this vast amount of corporate data has the advantage that it can be easily collected and analyzed through web crawling. This study therefore aims to collect and analyze information on job satisfaction provided by former and current employees over nine industries from Job Planet, which is one of the top job portal sites in South Korea, using the two-factor theory. In many previous studies, the motivating factors and hygiene factors of the 2-factor theory were tested for specific industries or employees using the survey method, and there is a limitation in that the number of samples is small [21–26]. Therefore, in this study, in order to supplement these limitations, a large amount of corporate data was collected from job portal sites to verify the motivational factors and hygiene factors of the 2-factor theory. In addition, this study aims to compare the differences in factors affecting job satisfaction for each of the nine industries and current and former employees. Through this, it can be said that this study supplemented the limitations of existing studies and secured the reliability of the research results.

This study is expected to be a case in which vast company data from job portal sites are used in human resource management as well as the 2-factor theory. In addition, if a company establishes a personnel management strategy that reflects the characteristics of the industry based on the results of this study, it can be expected to improve the job satisfaction of its members.

## 2. Research Background

### 2.1. Two-Factor Theory

The two-factor theory states that there are two independent sets of factors, which are motivation and hygiene factors, that affect one's job satisfaction and dissatisfaction, respectively [11]. Table 1 shows the classification of motivation and hygiene factors. Motivation factors are related to an employee's performance which reflect the intrinsic aspects of their jobs encompassing achievement, recognition, the job itself, responsibility, advancement, and possibility of growth. These factors motivate employees to feel satisfied of their job. While dissatisfaction will not occur even if such factors are not satisfied, the level of job satisfaction increases which induces a better work attitude considering that they are satisfied. On the other hand, hygiene factors are related to the environment which reflects the external aspects of work including company policy and administration, technical supervision, interpersonal relation, working conditions, salary, personal life, status, and job security. These factors imply environmental conditions that could alleviate the employees' dissatisfaction. It is worthwhile to note that the hygiene factors can lower dissatisfaction when satisfied but are incapable of generating satisfaction. Therefore, the opposite of satisfaction would be the none of satisfaction, while the opposite of dissatisfaction would be the none of dissatisfaction.

**Table 1.** Herzberg's motivation and hygiene factors [11].

| Motivation Factors | Hygiene Factors |
| :---: | :---: |
| Achievement | Company Policy & Administration |
| Recognition | Supervision·Technical |
| Work Itself | Interpersonal Relation |
| Responsibility | Working Conditions |
| Advancement | Salary |
| Possibility of Growth | Personal Life |
| | Status |
| | Job Security |

An introduction of the two-factor theory has aroused controversy among many studies either supporting or opposing its concept. Maidani [27] tested the two-factor theory on both employees working in public institutions and those working in private organizations, approving that motivation factors affect job satisfaction whereas hygiene factors do not, and thus supported the two-factor theory. On the other hand, Ewen [28] criticized the two-factor theory by confirming that extrinsic factors, such as the work environment, have a greater influence on job satisfaction and that job satisfaction and dissatisfaction factors can vary depending on organizational situations. Furthermore, researchers have criticized the two-factor theory for placing too much importance on satisfaction factors relative to job dissatisfaction factors [29,30]. However, many researchers have all agreed upon the two-factor theory for providing practical yet precise methods for job satisfaction analysis.

Despite a notion of controversy, the two-factor theory has also been used for determining job satisfaction and dissatisfaction among employees. After analyzing job satisfaction of 287 hotel staff based on the two-factor theory, Chitiris [31] argued that the hygiene factors from the two-factor theory would affect job satisfaction. Simons and Enz [32] suggested that hygiene factors, such as high wages and job security, in the service industry play more important roles than motivation factors. Using the two-factor theory, Derby-Davis [33] analyzed the relationship between job satisfaction and tenure intention of professors from the Department of Nursing and concluded that motivational factors positively affect the professors' tenure intention. On the other hand, Prasad Kotni and Karumuri [34] analyzed job satisfaction by applying the two-factor theory to 150 sales retailers, concluding that hygiene factors had a more significant effect on job satisfaction than motivation factors, which was inconsistent with the two-factor theory. Sobaih and Hasanein [35] applied the two-factor theory to five-star hotel employees residing in 10 different countries and demonstrated that motivation factors had a negative effect on job satisfaction, which can be interpreted as the cause of job dissatisfaction; however, they found hygiene factors to positively affect job satisfaction. These results suggest that the two-factor theory does not universally apply to all organizations as well as workers. Alrawahi et al. [10] analyzed job satisfaction of medical workers based on the two-factor theory, which yielded a result of health and safety, heavy workload, salary, promotion, recognition, and organizational policy to be the factors responsible for job dissatisfaction. On the other hand, they found that relationships with colleagues, relationships with leaders, and professional development to be the factors affecting job satisfaction, which confirmed that such results were inconsistent with the existing two-factor theory.

This study uses corporate review data spread over nine industries from Job Planet to identify motivation factors as well as hygiene factors affecting job satisfaction by each sector and determine whether there is a significant effect on job satisfaction through an application of the two-factor theory.

*2.2. Job Satisfaction*

There are several definitions suggested by different researchers on job satisfaction. Locke [36] defined job satisfaction as the emotional state that employees feel through job activities, whereas Saunders [37] defined it as one's interest or enthusiasm for achieving individual and organizational goals in a given environment. Osborn [38] defined job satisfaction as the degree to which an individual feels positive or negative about various aspects such as job tasks, working conditions, and peer relationships. Hoy and Miskel [39] defined job satisfaction as emotions that employees have with regard to a gap between their expected and actual career outputs. Robbins and Judge [40] defined job satisfaction as the overall attitude toward one's job, so that employees with high job satisfaction maintain a positive attitude whereas those with high job dissatisfaction maintain a negative attitude. Dessler [41] defined it as the emotional response to various aspects of the job. Having summarized the definitions of job satisfaction proposed by the previous studies, we are able to advocate job satisfaction as the emotions that employees retain throughout their work performance. According to Chiaburu et al. [42], employees who feel satisfied with their job tend to stay in the organization, whereas those who are dissatisfied are more likely to leave the organization. In other words, job satisfaction of employees has a significant impact on their turnover intention. An employee's turnover behavior is explained by one's desire to leave the organization if factors such as salary, promotion opportunity, the relationship with one's boss, responsibility, and autonomy provided by the organization are lower than their expectations [43]. To achieve an utmost corporate performance, it is important for employees to maintain high satisfaction towards their job [44].

As shown in Table 2, the constituent factors of job satisfaction are slightly different between researchers. Gruneberg [45] classified them into intrinsic and extrinsic factors; intrinsic factors are related to job performance which include job performance, autonomy and diversity, identity, as well as importance; whereas extrinsic factors are related to the work environment, such as working conditions, relationships with one's boss and colleagues, corporate culture, and wages. Huang et al. [19] argued that factors such as compensation/benefits, work–life balance, managers, and growth opportunities affect employees' job satisfaction. Ling [46] measured six factors that consisted of salary, status, job stability, recognition, a sense of belonging, and creativity to understand the overall job satisfaction of employees. Melián-González et al. [47] confirmed that satisfaction with salary, welfare, work–life balance, and the managers' leadership affects the overall job satisfaction. Saari and Judge [48] defined the work environment, type of work, and compensation as factors responsible for job satisfaction. Seashore and Taber [49] classified job satisfaction factors into environmental and personal factors. Environmental factors include job characteristics and the organizational environment, whereas personal factors include age, gender, and one's education level. Smith et al. [50] developed a Job Description Index (JDI) consisting of 72 questionnaires on five factors such as the job itself, salary, promotion, supervision, and peer relationships for measuring job satisfaction. According to Ting [51]'s findings, factors such as wage satisfaction, promotion opportunities, clarity of one's role, and peer relationships influence job satisfaction. Orgambídez-Ramos and de Almeida [52] claimed that personal factors (ability, competency, beliefs) and environmental factors (salary, promotion, relationship with boss and colleagues) create a huge impact on job satisfaction of employees. Wang et al. [53] ascertained that working conditions and job suitability as the factors affecting job satisfaction. Perera and John [54] suggested factors such as compensation, welfare, the job itself, and supervision for measuring employees' job satisfaction. In comparison with the two-factor theory, the constituent factors of job satisfaction presented by the previous studies do not show a significant difference.

**Table 2.** Key constructs of job satisfaction.

| Researchers | Constructs |
|---|---|
| [45] | Job Performance, Company Culture, Job Autonomy and Diversity, etc. |
| [19] | Compensation and Benefits, Manager, Work–life Balance, etc. |
| [46] | Salary, Status, Job Stability, Recognition, Sense of Belonging, etc. |
| [47] | Salary and Welfare, Work–life Balance, Management Leadership, etc. |
| [48] | Work Environment, Work Type, Compensation, etc. |
| [49] | Job Characteristics, Organizational Environment, Age, Gender, Education, etc. |
| [50] | Job Itself, Salary, Promotion, Supervision, Peer Relationships |
| [51] | Salary, Promotion Opportunities, Role Clarity, Peer Relationships |
| [52] | Personal Factors (Ability, Competency, Beliefs), Environmental Factors (Salary, Promotion, Relationship with Boss and Colleagues) |
| [53] | Working Conditions, Job Suitability |
| [54] | Compensation, Welfare, Job Itself, Supervision |

## 3. Research Model and Hypothesis Development

### 3.1. Research Model

In this study, we examined the impact of promotion opportunities/possibilities, welfare and salary, work–life balance, corporate culture, and management satisfaction on the employees' overall job satisfaction as shown in Figure 1. Here, we also wanted to check a potential difference in job satisfaction between current and former employees.

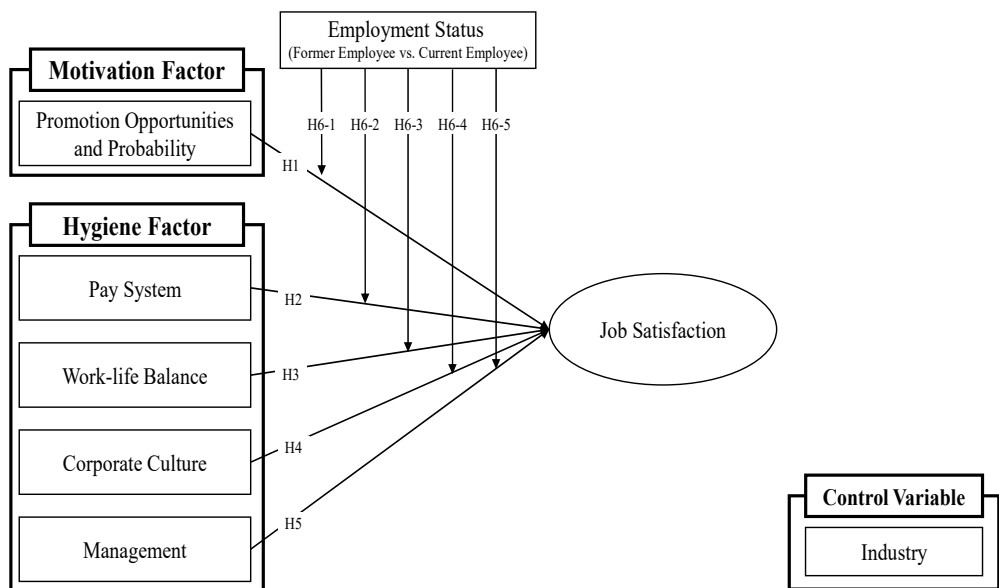

**Figure 1.** Research model.

Having referred to the constituent factors of job satisfaction defined in previous studies, we incorporated promotion opportunities/possibilities, pay system, work–life balance, corporate culture, and management as the main factors that are judged to be appropriate for determining job satisfaction of former and current employees from nine industries. Although these factors are also the ones used to indicate job satisfaction on the review page of Job Planet, we included them with our own definitions based on the two-factor theory. In the next section, we present research hypotheses to scrutinize how sub-factors from the motivation and hygiene factors would affect employees' job satisfaction after a close examination of the extant literature.

### 3.2. Research Hypothesis

### 3.2.1. The Relationship between Promotion Opportunities/Possibilities and Job Satisfaction

Promotion is part of the personnel system within an organization which plays a role in enhancing employees' performance by promoting one's job stability as well as encouraging motivation for his or her job. Corporate support for promotion and self-development improves job satisfaction and organizational productivity by relieving employees' career-related anxiety [55]. In other words, as employees perceive higher possibility of promotion, their overall job satisfaction rises as well as the sense of stability [56]. According to Razak et al. [57], satisfaction with regard to promotion affects the overall job satisfaction; the more systematic the promotion system, the higher the job satisfaction and work performance. Paarsch and Shearer [58] proposed that the promoted employees are delighted with their corporate promotion system and its opportunities while carrying high expectations for another promotion. However, ones dissatisfied with the promotion opportunities and system are likely to quit their job [59], whereas ones with more opportunities for promotion demonstrate higher job satisfaction [60]. Therefore, it is a factor that is related to the job itself performed by organizational members and motivates them to create higher performance, which can be considered as a motivating factor. In this study, promotion opportunities and possibilities were defined as a part of the personnel system within an organization that promotes one's job stability, inspires motivation for work, and improves work performance, which lead to our first hypothesis.

**H1.** *Satisfaction with promotion opportunities and possibilities will significantly affect one's job satisfaction.*

### 3.2.2. The Relationship between a Pay System and Job Satisfaction

A pay system is an important factor in job satisfaction that directly affects an individual's economic activities and the cost of labor [61,62]. This indicates that the welfare and pay system are essential for preventing an unprecedented departure of employees as well as promoting work performance [63–65]. If employees are not provided with an appropriate amount of salary or welfare benefits relative to their workload, they will feel dissatisfied with their job as they adopt negative attitudes and emotions [66,67]. Dissatisfaction with the pay system and welfare greatly affects employees' job attitude and their voluntary turnover intention, which leads to unproductive human resource performance of the company [68]. Brown [69] made a remark that an increase in an employee's salary has a significant effect on his or her job satisfaction which would also increase their corporate stability. According to Machová et al. [70], they listed seven types of motivation tools and financial incentives (salary increases and bonuses) that were the most effective motivation tool for the employees. Therefore, if an appropriate salary is provided, an environment is created in which the members of the organization can concentrate on their jobs without feeling uncomfortable in economic activities. This is related to the environment outside of the job performed by organizational members and can be considered to be included in the hygiene factor. In this study, we established the following hypothesis by defining one's welfare and salary as the cost of labor which directly influences an individual's economic activities.

**H2.** *Satisfaction with the pay system will have a significant effect on one's job satisfaction.*

### 3.2.3. The Relationship between Work–Life Balance Factors and Job Satisfaction

Work–life balance refers to a range of satisfaction felt by employees through a balance of their work and all the other aspects of life [71,72]. Recently, companies have implemented systems such as flexible working hours and voluntary commuting to improve the employees' satisfaction with regard to their work–life balance [73]. Moreover, work–life balance not only increases the employees' quality of life and job satisfaction, but also positively affects the organizational productivity [74]. When employees are given autonomy

to work anytime, anywhere in a personal space other than their office, their overall job satisfaction turns out to be higher [75,76]. In addition, systems such as flexible working hours, in-house systems that allow employees to use annual leave voluntarily, and guarantees of working hours enhance the employees' job satisfaction [77,78]. However, since the traditional working time system does not consider the differences between individuals' capability, inefficiencies occur in work performance as, for instance, one may waste his or her time staying seated even after completing one's work [79]. This work–life balance provides satisfaction in the personal life of organizational members, thereby increasing the productivity of the company. Therefore, the work–life balance corresponds to the hygiene factor related to the surrounding environment of organizational members. Therefore, in this study, work–life balance is defined as a state in which physical or psychological activities are appropriately distributed in various aspects of the employees' life where one feels satisfaction with his or her life accordingly, which leads to the following hypothesis.

**H3.** *Satisfaction with work–life balance will have a significant effect on one's job satisfaction.*

### 3.2.4. The Relationship between Corporate Culture and Job Satisfaction

Corporate culture encompasses the beliefs, values, norms, and philosophies formed through the interactions between employees, which acts as an informal guideline that determines the behavior of individuals within the organization [80,81]. If corporate culture and the employees' values of the organization are consistent, they will feel comfortable towards their work environment that ultimately enhances their job satisfaction [82]. In the end, a friendly corporate culture positively affects employees' job satisfaction, whereas a hierarchical culture creates a negative impact [83]. Furthermore, a horizontal corporate culture, which can be explained by a smooth communication between corporate members and a participatory environment, affects employees' job satisfaction [84]. In other words, the corporate culture determines the job satisfaction of organizational members by the organizational environment, such as organizational values and mindset, so it can be considered as a hygiene factor. In this study, corporate culture was defined as the informal guidelines that determine the beliefs, values, norms, and philosophies formed through the interactions among employees and their individual behaviors within the organization, which is hypothesized as follows.

**H4.** *Satisfaction with corporate culture will have a significant effect on one's job satisfaction.*

### 3.2.5. The Relationship between Management and Job Satisfaction

Management mainly refers to the relationship between employees and the board of directors regarding how well the managers understand employees' role and their tasks [85]. According to Brown et al. [86], the organization's efficiency is improved when a trust between employees and the management is well formed; however, mistreated employees, mainly due to ignorant, violent leaders, generally had low job satisfaction. These negative influences increase personal anxiety and emotional fatigue and thus lower the employees' job performance [87]. Therefore, the attitude and behavior of the management toward employees directly affect their morale and job satisfaction [88]. According to Roethlisberger and Dickson [89], the employees' attitude would change in accordance with the change in the attitude and behaviors of the management. With favorable methods of supervision and generous attitudes of the supervisors, employees showed higher job satisfaction, conscientiousness, as well as loyalty. In other words, the relationship with the management determines the performance, loyalty, and job satisfaction of organizational members, which can be viewed as related to the surrounding environment than the job of the organizational member, so it can be said to be included in the hygiene factor. Thus, we established the following hypothesis after defining the management as the relationship between the directors and employees, or how well the former understands the employees' tasks of work.

**H5.** *Satisfaction with the management will have a significant effect on one's overall job satisfaction.*

### 3.2.6. Moderating Effect of Former or Current Employees

Mobley [90] defined a former employee as one who quitted his or her job from the company that used to provide one financial compensation; a former employee has either changed or quitted the current job to move onto another. Previous studies were mainly conducted with regard to one's turnover intention [91–94]. However, even though the employees' behaviors can be predicted through their intentions, there are cases where intentions do not necessarily lead to actual actions [95]. In other words, it is challenging to put intention and action on the same line with the above reason. Nevertheless, not many studies have analyzed job satisfaction of both former and current employees. Therefore, in this study, we established the following hypotheses:

**H6.1.** *The employment status (former employees vs. current employees) moderates the relationship between satisfaction with promotion opportunities and possibilities on job satisfaction.*

**H6.2.** *The employment status (former employees vs. current employees) moderates the relationship between satisfaction with the pay system on job satisfaction.*

**H6.3.** *The employment status (former employees vs. current employees) moderates the relationship between satisfaction with work–life balance on job satisfaction.*

**H6.4.** *The employment status (former employees vs. current employees) moderates the relationship between satisfaction with corporate culture on job satisfaction.*

**H6.5.** *The employment status (former employees vs. current employees) moderates the relationship between satisfaction with management on job satisfaction.*

### 3.3. Dataset

In this study, we used a web-crawling technique to collect data from Job Planet which contains information of companies from 2014 to 2021 over nine industries including IT web communication, construction, education, media design, service, trade/transport, banking and finance, pharmaceutical/medical welfare, and chemical manufacturing. Such information is shown in Figure 2, which includes a 5-point scale that measures job satisfaction, promotion opportunities and possibilities, pay system, work–life balance, corporate culture, and the ratings of the management. A description of each variable is given in Table 3.

**Table 3.** Data description.

| Variable | Description |
| --- | --- |
| Industry | Nine industries including IT web communication, construction, education, media design, service, trade/transport, banking and finance, pharmaceutical/medical welfare, and chemical manufacturing |
| Employment Status | Former or Current employee (binary scale) |
| Job Satisfaction (JS) | Overall job satisfaction (5-point scale) |
| Promotion Opportunities and Probability (PO) | Satisfaction with HR system within the organization (5-point scale) |
| Pay System (PS) | Satisfaction with pay system or wages within the organization (5-point scale) |
| Work–life Balance (WB) | Satisfaction with balance of work and life (5-point scale) |
| Corporate Culture (CC) | Satisfaction with the culture established within the organization (5-point scale) |
| Management (MG) | Satisfaction with relationship with management or boss's treatment (5-point scale) |

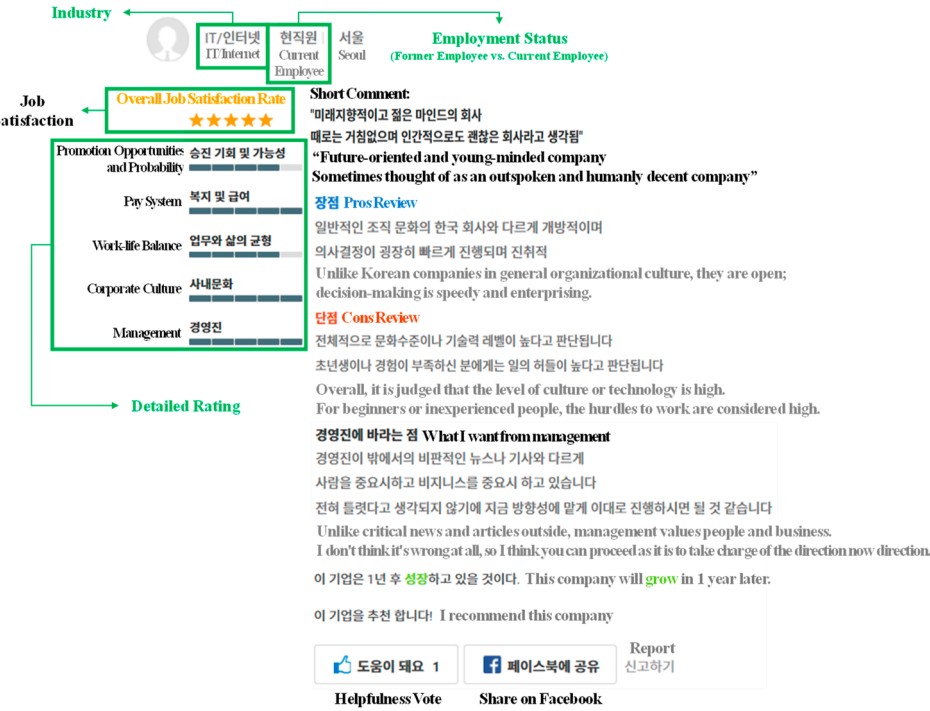

**Figure 2.** Data collection.

## 4. Results

### 4.1. Descriptive Statistics

We collected information regarding job satisfaction, promotion opportunities and possibilities, welfare/salary, work–life balance, corporate culture, and the management on a 5-point scale given by current and former employees over nine industries from 2014 to 2021 on Job Planet. We used a total of 355,199 data, as shown in Table 4.

**Table 4.** Descriptive statistics about former and current employees.

| Former Employees | Current Employees | Total |
| --- | --- | --- |
| 224,731 | 130,486 | 355,199 |

Table 5 shows the descriptive statistics for each variable. The average of job satisfaction was about 3.0, which can be seen that most current and former employees tend to give a rating of 3 points. However, the rating for the management was 2.68, indicating that satisfaction with the management was somewhat lower than that of other factors.

**Table 5.** Descriptive statistics about job satisfaction rating and detailed rating.

| Variable | Mean | SD | N |
| --- | --- | --- | --- |
| JS | 3.17 | 1.030 | 355,199 |
| PO | 2.98 | 1.019 | 355,199 |
| PS | 3.09 | 1.109 | 355,199 |
| WB | 3.03 | 1.226 | 355,199 |
| CC | 3.06 | 1.144 | 355,199 |
| MG | 2.68 | 1.080 | 355,199 |

The results of correlation analysis between variables are shown in Table 6, and it can be confirmed that there is a positive correlation between job satisfaction and each variable. Additionally, since the correlation coefficients between the variables are all less than 0.9, it can be seen that multicollinearity between the variables does not exist [96].

**Table 6.** Correlation analysis.

|  | **JS** | **PO** | **PS** | **WB** | **CC** | **MG** |
|---|---|---|---|---|---|---|
| JS | 1 |  |  |  |  |  |
| PO | 0.531 ** | 1 |  |  |  |  |
| PS | 0.594 ** | 0.456 ** | 1 |  |  |  |
| WB | 0.561 ** | 0.305 ** | 0.379 ** | 1 |  |  |
| CC | 0.626 ** | 0.434 ** | 0.414 ** | 0.558 ** | 1 |  |
| MG | 0.626 ** | 0.484 ** | 0.493 ** | 0.461 ** | 0.602 ** | 1 |

** $p < 0.01$.

*4.2. Analysis over All Industries*

We performed path analysis to explore factors affecting job satisfaction. Table 7 shows the results for all industries controlled by an industry. Promotion opportunities and possibilities ($\beta = 0.163$, $p < 0.001$), the pay system ($\beta = 0.255$, $p < 0.001$), work–life balance ($\beta = 0.204$, $p < 0.001$), corporate culture ($\beta = 0.218$, $p < 0.001$) and the management ($\beta = 0.195$, $p < 0.001$) were all confirmed to have a significant positive (+) effect on the dependent variable being job satisfaction. It is therefore logical that these factors are responsible for making people feel job satisfaction. Therefore, hypotheses H1, H2, H3, H4, and H5 were accepted.

**Table 7.** Analysis of determinants of job satisfaction (all industries).

| Path | B | β | S.E. |
|---|---|---|---|
| Ref: IT Web Communication |  |  |  |
| Construction→JS | 0.065 *** | 0.014 *** | 0.006 |
| Education→JS | 0.028 *** | 0.006 *** | 0.006 |
| Media Design→JS | 0.043 *** | 0.011 *** | 0.005 |
| Service→JS | 0.013 ** | 0.004 ** | 0.004 |
| Trade/Transport→S | 0.032 *** | 0.011 *** | 0.004 |
| Banking and Finance→JS | 0.006 | 0.002 | 0.005 |
| Pharmaceutical/Medical Welfare→JS | −0.013 ** | −0.003 ** | 0.005 |
| Chemical Manufacturing→JS | 0.034 *** | 0.014 *** | 0.004 |
| PO→JS | 0.165 *** | 0.163 *** | 0.001 |
| PS→JS | 0.237 *** | 0.255 *** | 0.001 |
| WB→JS | 0.171 *** | 0.204 *** | 0.001 |
| CC→JS | 0.196 *** | 0.218 *** | 0.001 |
| MG→JS | 0.186 *** | 0.195 *** | 0.001 |
| SMC |  | 0.612 |  |

** $p < 0.01$, *** $p < 0.001$, SMC = Squared Multiple Correlation.

Table 8 shows the results of examining the differences in job satisfaction according to the employment status. Promotion opportunities and possibilities for both former and current employees (former employees: $\beta = 0.141$, $p < 0.001$; current employees: $\beta = 0.192$, $p < 0.001$), the pay system (all employees: $\beta = 0.251$, $p < 0.001$; current employees: $\beta = 0.270$, $p < 0.001$), work–life balance (all employees: $\beta = 0.193$, $p < 0.001$; current employees: $\beta = 0.210$, $p < 0.001$), corporate culture (all employees: $\beta = 0.226$; $p < 0.001$, current employees: $\beta = 0.201$, $p < 0.001$) and the management (all employees: $\beta = 0.206$, $p < 0.001$; current employees: $\beta = 0.188$, $p < 0.001$) had a positive (+) effect on job satisfaction. After examining the path difference between such former and current employees, the effects of promotion opportunities and possibilities, the pay system, and work–life balance on job satisfaction were greater among current employees than former employees. On the other hand, the influence of corporate culture and the management on job satisfaction was more significant for former employees than current employees. As a result, hypotheses H6-1, H6-2, H6-3, H6-4, and H6-5 were all accepted.

**Table 8.** Multi-group path analysis of motivation and hygiene factors for job satisfaction (all industries).

| Path | Former Employees | | | Current Employees | | | Path Differences between Groups |
|---|---|---|---|---|---|---|---|
| | **B** | **β** | **S.E.** | **B** | **β** | **S.E.** | |
| Ref: IT Web Communication | | | | | | | |
| Construction→JS | 0.055 | 0.011 | 0.007 | 0.075 *** | 0.017 *** | 0.008 | −1.783 |
| Education→JS | 0.027 | 0.006 | 0.007 | 0.064 *** | 0.012 *** | 0.01 | −3.143 ** |
| Media Design→JS | 0.055 *** | 0.016 *** | 0.006 | 0.052 *** | 0.012 *** | 0.008 | 0.347 *** |
| Service→JS | 0.034 *** | 0.012 *** | 0.005 | 0.006 | 0.002 | 0.007 | 3.148 ** |
| Trade/Transport→JS | 0.045 *** | 0.016 *** | 0.005 | 0.028 *** | 0.009 *** | 0.006 | 2.006 * |
| Banking and Finance→JS | 0.006 *** | 0.002 *** | 0.006 | 0.012 | 0.004 | 0.007 | −0.732 |
| Pharmaceutical/Medical Welfare→JS | −0.012 *** | −0.003 *** | 0.006 | −0.017 | −0.004 | 0.007 | 0.527 |
| Chemical Manufacturing→JS | 0.049 *** | 0.019 *** | 0.005 | 0.003 | 0.001 | 0.005 | 6.39 *** |
| PO→JS | 0.143 *** | 0.141 *** | 0.002 | 0.194 *** | 0.192 *** | 0.002 | −19.138 *** |
| PS→JS | 0.232 *** | 0.251 *** | 0.002 | 0.249 *** | 0.27 *** | 0.002 | −6.799 *** |
| WB→JS | 0.164 *** | 0.193 *** | 0.001 | 0.173 *** | 0.21 *** | 0.002 | −4.054 *** |
| CC→JS | 0.205 *** | 0.226 *** | 0.002 | 0.178 *** | 0.201 *** | 0.002 | 9.813 *** |
| MG→JS | 0.199 *** | 0.206 *** | 0.002 | 0.173 *** | 0.188 *** | 0.002 | 9.604 *** |
| SMC | | 0.600 | | | 0.625 | | |

* $p < 0.05$, ** $p < 0.01$, *** $p < 0.001$, SMC = Squared Multiple Correlation.

### 4.3. Analysis of an Individual Industry

Table 9 shows the path analysis results for job satisfaction for each industry. Motivation and hygiene factors that affect job satisfaction over nine industries are as follows. Therefore, hypotheses H1, H2, H3, H4, and H5 were adopted for all nine industries.

**Table 9.** Analysis of determinants of job satisfaction (for each industry).

| Path | IT Web Communication | Construction | Education | Media Design | Service |
|---|---|---|---|---|---|
| PO→JS | 0.193 *** | 0.161 *** | 0.124 *** | 0.140 *** | 0.150 *** |
| PS→JS | 0.250 *** | 0.270 *** | 0.262 *** | 0.245 *** | 0.253 *** |
| WB→JS | 0.170 *** | 0.192 *** | 0.219 *** | 0.169 *** | 0.180 *** |
| CC→JS | 0.218 *** | 0.221 *** | 0.202 *** | 0.247 *** | 0.204 *** |
| MG→JS | 0.206 *** | 0.188 *** | 0.225 *** | 0.230 *** | 0.211 *** |
| SMC | 0.602 | 0.631 | 0.657 | 0.609 | 0.564 |
| | **Trade/Transport** | **Banking and Finance** | **Pharmaceutical/Medical Welfare** | **Chemical Manufacturing** | |
| PO→JS | 0.158 *** | 0.167 *** | 0.178 *** | 0.169 *** | |
| PS→JS | 0.227 *** | 0.256 *** | 0.257 *** | 0.263 *** | |
| WB→JS | 0.223 *** | 0.226 *** | 0.189 *** | 0.229 *** | |
| CC→JS | 0.218 *** | 0.223 *** | 0.233 *** | 0.206 *** | |
| MG→JS | 0.201 *** | 0.177 *** | 0.193 *** | 0.168 *** | |
| SMC | 0.582 | 0.619 | 0.612 | 0.628 | |

*** $p < 0.001$, SMC = Squared Multiple Correlation.

The results of examining the moderating effect of the employment status between former and current employees are shown in Table 10. Since different results were derived for each industry, the hypothesis on the moderating effect of employment status on current and former employees was partially adopted. Overall, in terms of satisfaction with promotion opportunities and possibilities, pay system, and work–life balance, the satisfaction of current employees was higher than that of former employees. However, in terms of satisfaction with the corporate culture and management, it was found that the satisfaction of former employees was higher than that of current employees.

**Table 10.** Multi-group path analysis of job satisfaction (for each industry).

| Path | PO→JS | | | PS→JS | | | WB→JS | | |
|---|---|---|---|---|---|---|---|---|---|
| | Former Employee | Current Employee | Coefficient | Former Employee | Current Employee | Coefficient | Former Employee | Current Employee | Coefficient |
| IT Web Communication | 0.177 *** | 0.206 *** | −3.744 *** | 0.239 *** | 0.273 *** | −4.397 *** | 0.164 *** | 0.164 *** | 0.554 |
| Construction | 0.128 *** | 0.195 *** | −4.966 *** | 0.264 *** | 0.284 *** | −1.300 | 0.192 *** | 0.181 *** | 1.497 |
| Education | 0.098 *** | 0.174 *** | −6.225 *** | 0.258 *** | 0.279 *** | −1.320 | 0.228 *** | 0.197 *** | 2.075 * |
| Media Design | 0.119 *** | 0.184 *** | −5.761 *** | 0.253 *** | 0.236 *** | 2.051 * | 0.161 *** | 0.176 *** | −1.071 |
| Service | 0.138 *** | 0.179 *** | −4.654 *** | 0.248 *** | 0.275 *** | −3.423 *** | 0.173 *** | 0.186 *** | −1.098 |
| Trade/Transportation | 0.139 *** | 0.187 *** | −6.203 *** | 0.225 *** | 0.241 *** | −2.466 * | 0.204 *** | 0.241 *** | −2.775 ** |
| Banking and Finance | 0.141 *** | 0.201 *** | −6.720 *** | 0.247 *** | 0.274 *** | −2.188 * | 0.208 *** | 0.239 *** | −3.270 ** |
| Pharmaceutical & Medical Welfare | 0.168 *** | 0.184 *** | −1.225 | 0.255 *** | 0.267 *** | −1.402 | 0.175 *** | 0.202 *** | −2.294 ** |
| Chemical Manufacturing | 0.144 *** | 0.194 *** | −10.170 *** | 0.256 *** | 0.274 *** | −2.785 ** | 0.219 *** | 0.234 *** | −1.398 |

| Path | CC→JS | | | MG→JS | | | SMC | | |
|---|---|---|---|---|---|---|---|---|---|
| | Former Employee | Current Employee | Coefficient | Former Employee | Current Employee | Coefficient | Former Employee | Current Employee | |
| IT Web Communication | 0.224 *** | 0.206 *** | 2.977 ** | 0.218 *** | 0.204 *** | 3.981 *** | 0.592 | 0.608 | |
| Construction | 0.223 *** | 0.214 *** | 1.421 | 0.207 *** | 0.175 *** | 3.718 *** | 0.621 | 0.636 | |
| Education | 0.203 *** | 0.191 *** | 0.415 | 0.228 *** | 0.216 *** | 1.567 | 0.637 | 0.676 | |
| Media Design | 0.253 *** | 0.224 *** | 3.163 ** | 0.231 *** | 0.235 *** | 1.439 | 0.601 | 0.616 | |
| Service | 0.204 *** | 0.200 *** | 0.382 | 0.208 *** | 0.190 *** | 1.978 * | 0.553 | 0.597 | |
| Trade/Transportation | 0.224 *** | 0.200 *** | 3.128 ** | 0.212 *** | 0.194 *** | 3.134 ** | 0.574 | 0.593 | |
| Banking and Finance | 0.241 *** | 0.197 *** | 4.697 *** | 0.193 *** | 0.164 *** | 3.929 *** | 0.612 | 0.621 | |
| Pharmaceutical & Medical Welfare | 0.253 *** | 0.201 *** | 5.117 *** | 0.191 *** | 0.201 *** | 0.404 | 0.596 | 0.618 | |
| Chemical Manufacturing | 0.219 *** | 0.189 *** | 6.052 *** | 0.177 *** | 0.165 *** | 3.403 *** | 0.618 | 0.640 | |

\* $p < 0.05$, \*\* $p < 0.01$, \*\*\* $p < 0.001$, SMC = Squared Multiple Correlation.

## 5. Discussion

To achieve the objectives of our study, we collected and analyzed information on job satisfaction provided by former and current employees over nine industries from job portal sites, using the two-factor theory. Therefore, the findings of this study are as follows: First, we confirmed that the factors for promotion opportunities and possibilities had a positive (+) effect on job satisfaction over all industries. This coincides with the results of previous studies [58,61] which support this study's results. It is therefore logical to claim that the overall job satisfaction gets higher as employees are more satisfied with promotion opportunities and possibilities, which are mainly determined by whether their promotion and personnel evaluation system is reasonable [97,98]. Furthermore, if employees perceive that the organization's promotion and evaluation system is fair, they will accept the system's results even if they are negative and still feel satisfied with their job [99]. Therefore, we highly suggest companies to establish specific criteria for promotion and personnel evaluation that secures fairness so that employees can accept the results from such systems.

Second, we approved that the factors on welfare and salary had a positive (+) effect on job satisfaction over nine industries, which demonstrates the same results as ones of previous studies [100–104]; the higher the satisfaction with welfare and salary, the higher the employees' overall job satisfaction. In fact, the welfare and salary provided by companies indicate the compensation for one's labor activities [105,106] which directly affects the employees' economic activities [62,63]. An enhancement in corporate productivity comes from employees' individual work efficiency as well as a low turnover rate, which can be achieved through a provision of diverse welfare types and higher salaries [107,108]. In other words, if a company establishes a suitable welfare and salary system considering its internal circumstances, employees' satisfaction may increase which can also generate higher corporate performance [109]. For these reasons, companies ought to provide their employees with an effectively designed welfare and salary system after adequate consideration of their corporate circumstances as well as the industry characteristics.

Third, we confirmed that the factors for work–life balance had a positive (+) effect on job satisfaction over all nine industries, which matches with the results of previous

studies [75–78]. Therefore, the higher satisfaction with the work–life balance of the employees, the higher their overall job satisfaction. In other words, when the requirements for work–life balance are satisfied, employees feel satisfied with their job that positively affects corporate performance [78]. If a company prepares a work–life balance system, it will alleviate one's concern of maintaining his role between the company and household that may reduce unnecessary energy in performing various duties [110]. For instance, job satisfaction increases when employees can adjust working hours and personal schedules with discretion [74]. Therefore, this study suggests companies to promote a work–life balance policy suitable for the corporate circumstances and allow employees to adjust their working hours and schedules freely.

Fourth, we found that the factors regarding corporate culture had a positive (+) effect on job satisfaction in all nine industries, which coincides with the results of previous studies [111–114]. This indicates that the higher the employees' satisfaction with corporate culture, the higher the overall job satisfaction. Moreover, the higher the degree of connection between corporate culture and preferred culture of employees, the higher the job satisfaction as well as the positive intention to work [82]. It is evident that favorable relationships and culture within the organization arouse more positive job behaviors, such as cooperation and altruism among employees, and ultimately the whole corporate performance [115,116]. Therefore, it is necessary to create a lateral culture with respect among all organizational members to improve higher satisfaction on corporate culture.

Fifth, we figured that the factors of the management had a positive (+) effect on job satisfaction over all nine industries, which was identical to the results of previous studies [117–119]. The higher the satisfaction of the employees with the management, the higher the overall job satisfaction. Satisfaction with the corporate management comes from their job performance and technical ability, as they are the ones who make decisions and thereby have a considerable influence on employees [120,121]. Management support refers to giving interest and assistance to duties performed by employees [122]. Such a support would include for instance the management providing advice and information on employees' performance, which positively affects the overall job satisfaction [123]. Furthermore, one of the essential factors in the management's satisfaction is a technical ability on how well the managers understand the employees' tasks [122]. When the management shows an excellent ability in work performance, employees' satisfaction would increase, which positively affects the organization's productivity as well [85]. Therefore, companies ought to evaluate the management's performance with fairness and objectivity and create a trustworthy environment among employees and the management.

Lastly, the results of examining the moderating effect of the employment status between former and current employees turned out to be different for each industry, while the current employees were generally more satisfied than the former employees. It can be asserted that the higher the job satisfaction, the higher the degree of intention to stay in a current organization [124,125], supported by previous studies' results. However, for certain industries including IT web communication, media design, trade/transport, banking and finance, pharmaceutical/medical welfare, and chemical manufacturing, the satisfaction of former employees was higher than that of current employees in corporate culture factors. In addition, in the case of some industries, such as IT web communication, construction, service, trade/transport, and banking/finance, and manufacturing and chemical industries, it was found that the satisfaction of former employees was higher than that of current employees in management factors. It is therefore necessary to prepare fundamental policies and systems that can enhance not only employees' job satisfaction, but also their willingness to stay in the organization.

## 6. Conclusions

### 6.1. Summary

In this study, we carefully examined motivation and hygiene factors influencing the overall job satisfaction of both former and current employees throughout nine different

industries. We collected 355,199 data from Job Planet between 2014 and 2021 with regard to job satisfaction, promotion opportunities and possibilities, the pay system, work–life balance, corporate culture, and the management to perform path analysis. Therefore, as a result of analyzing all industries, it was found that motivation and hygiene factors had a positive (+) effect on job satisfaction, so the hypotheses H1, H2, H3, H4, and H5 were adopted. Additionally, since the moderating effect of both former and current employees was also significant, the hypotheses H6.1, H.6.2, H6.3, H6.4, and H6.5 were adopted. As a result of analysis by nine industries, it was found that motivation and hygiene factors in all nine industries had a positive (+) effect on job satisfaction, so the hypotheses H1, H2, H3, H4, and H5 were adopted. Additionally, since the moderating effect of current and former employees was different for each industry, the hypothesis was partially adopted.

*6.2. Academic and Practical Implications*

Here, the academic implications of this study follow. First, we have acknowledged a limitation in collecting extensive amounts of data through the existing methods of survey and interview. Therefore, this study is meaningful in a way that we could secure the representative sample after successfully obtaining 350,000 data scraped from Job Planet which were compared and analyzed over nine industries. If questionnaires and interviews were to be further developed based on the results of this study, more objective data may be rendered in the future.

Second, it is meaningful to analyze the role of the employment status as a moderating variable on job satisfaction using Herzberg's two-factor theory. Previous studies mainly focused on the turnover intention of current employees. However, since one's intention does not necessarily lead to an action, it is unreasonable to regard the intention and action as the same [95]. This study is expected to contribute to research on employees' turnover intention and their willingness-to-stay intention in the future as we conducted analysis on former and current employees who changed their jobs in real life.

Finally, meaningful results were derived by analyzing the job satisfaction of all and current employees using a vast amount of corporate data. This is considered to be an example of how to use big data analysis and its possibilities in the field of personnel management. In addition, the existing survey method has limitations in that the number of samples is limited and only specific industries and companies are analyzed. However, since this study analyzed the company information written by employees in various industries and companies, it can be said that the limitations of existing studies were supplemented, and the reliability of the research results was improved.

The practical implications of this study are as follows. First, we suggested implications for each determinant of job satisfaction of which employees feel satisfied. Through these implications, companies are expected to contribute to strategies for securing and maintaining human resources or establish job training programs to improve employees' job satisfaction.

Second, many job seekers recently show more trust and interest towards the information provided by job portal websites more than one directly provided by companies [126]. As smartphones have become more prevalent, there has been an exponential increase in the number of employment-related community users, which has generated even more diverse information. Thus, we collected and analyzed a large amount of data provided by the employment portal site that rendered significant results. In the end, companies need to systematically analyze their employees' job satisfaction through the collection of information from job portal sites or communities.

Third, unlike previous studies, it was found that the pay system, work–life balance, company culture, and management, which are hygiene factors, affect job satisfaction. Therefore, based on the results presented in this study, companies need to establish a systematic pay system in consideration of the workload and work intensity of organizational members and introduce flexible working systems and autonomous commuting systems so that work and life can be harmonized. In addition, if an internal culture preferred by the members of

the organization is formed and the management accurately grasps the work of the members of the organization and maintains a smooth relationship, performance can be expected to motivate the members of the organization and increase their job satisfaction.

Finally, in this study, factors affecting job satisfaction were analyzed by nine industries. Therefore, if a HR management strategy that reflects the characteristics of the industry is established, job satisfaction can be expected to improve more than the existing HR management strategy.

### 6.3. Limitations and Future Research

Despite having several implications, this study has the following limitations. First, we could only collect and analyze information that were quantitative. Since job portal sites also provide additional information such as reviews of companies, size of companies, and sales, the future studies could incorporate such information to propose more specific results.

Second, this study collected and analyzed data from 2014 to 2021. Since we did not explore the collected data in a seasonal manner, there is a limitation of neglecting the possible influence of events occurring at a specific time. Therefore, it is essential to analyze such data by period to comprehend the change in events in the future.

Lastly, this study carried out analysis on both the former and current employees. However, we did not consider the sociodemographic characteristics of the employees such as age, gender, and their education level. Considering extant literature suggesting the difference in results on job satisfaction by demographic characteristics [127,128], it is also necessary to conduct a detailed analysis that fully incorporates demographic characteristics of subjects in the future.

**Author Contributions:** Conceptualization, B.L., C.L., I.C. and J.K.; methodology, B.L.; software, B.L.; validation, B.L., C.L., I.C. and J.K.; formal analysis, J.K.; investigation, B.L.; resources, C.L.; data curation, I.C.; writing—original draft preparation, B.L.; writing—review and editing, I.C.; visualization, I.C.; supervision, C.L.; project administration, J.K. All authors have read and agreed to the published version of the manuscript.

**Funding:** This research received no external funding.

**Institutional Review Board Statement:** Not applicable.

**Informed Consent Statement:** Not applicable.

**Data Availability Statement:** Not applicable.

**Acknowledgments:** This research was supported by the Industrial Technology Innovation Program (20009050) and the Ministry of Trade, Industry & Energy (MOTIE, Korea).

**Conflicts of Interest:** The authors declare no conflict of interest.

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
