# Peer review of "Analyzing Determinants of Job Satisfaction Based on Two-Factor Theory"

_sustainability, doi:10.3390/su141912557_

Round 1
Reviewer 1 Report
I would like to first congratulate the authors for their work in this paper. I found some issues that need to be addressed:
The title does not match the variables that the researcher brings to the test. The two factors should be a theory that researchers want to use to explain the model and phenomenon of this study. Can the researcher adjust or revise the title that links with variables (optional)?
In the last paragraph should be added the research questions for making clear what you want to study.
There are many studies that have been shown the significance of factors (especially two factors) that affect employee satisfaction. It is advised to identify what is originality from your work in the introduction part.
Regarding the research methodology, there is needed to explain the instrument of this study come from (give an example or show all the questions in the appendix) and validate the instrument before collecting data. How many questions that you use? What kind of statistics are the employee in this study?
The discussion needs to be further enriched, based on existing literature in the paper, with more insight into the importance of the findings in this study. This will make the paper more compelling for its readers and make the importance of the findings stand out more. Also, a summary of the findings in view of the hypotheses needs to be added.
Author Response
Thank you for the constructive comments to help revise the manuscript.
Please see the attachment.

Reviewer 2 Report
Dear Authors,
find my comments attached! Great paper, only some "design" issues need to be solved!
All the best!

Author Response

(The authors gave the same response as above.)

Reviewer 3 Report
Dear Authors
It was with great pleasure that I reviewed your manuscript. The subject matter is of great importance.
However, I have some (few) considerations to make:
I would write hypotheses 6.1, 6.2, 6.3, 6.4, e 6.5 differently.
Instead, "The effect of satisfaction with promotion opportunities and possibilities on job satisfaction will vary depending on the employment status (former employees vs. current employees)." I would write, " The employment status (former employees vs. current employees) moderates the relationship between satisfaction with promotion opportunities and possibilities on job satisfaction ".
In your results, you did not put the coefficient of determination values (R2). It would be important that you do so.
Your study put the practical implications and limitations in your conclusions. It would be better if you separated them into a subchapter.
My Best Regards
Author Response

(The authors gave the same response as above.)
